# Metagenomic Analysis of Suansun, a Traditional Chinese Unsalted Fermented Food

**Yaping Hu** [1,†] **, Xiaodong Chen** [2,†] **, Jie Zhou** [1] **, Wenxuan Jing** [1] **and Qirong Guo** [1,3,*]

1 Co-Innovation Center for Sustainable Forestry in Southern China, Nanjing Forestry University, Nanjing 210037, China; huyaping@njfu.edu.cn (Y.H.); jiez0305@gmail.com (J.Z.); daxuan@njfu.edu.cn (W.J.)
2 East China Inventory and Planning Institute, National Forestry and Grassland Administration, Hangzhou 310000, China; 13161796388@163.com
3 International Center of Bamboo and Rattan, Beijing 100102, China
* Correspondence: qrguo@njfu.edu.cn; Tel.: +86-025-8542-8779
† These authors contributed equally to this work.

**Abstract:** Suansun, made from fresh bamboo shoots fermented without salt, is a traditional food in China's southern region and is popular for its nutritious and unique flavor. To comprehensively understand the microbial species and characteristics of suansun, Illumina HiSeq metagenomic sequencing technology was used to sequence suansun's fermentation broth obtained from six traditional producing areas in southern China, and the microbial community structure, diversity, and functional genes were analyzed. A total of 8 phyla, 16 classes, 30 orders, 63 families, 92 genera, and 156 species of microorganisms were identified in the suansun samples, with *Lactiplantibacillus* predominating, accounting for up to 81% of the species, among which 12 species, including *Lactiplantibacillus plantarum*, were the main species. A total of 12,751 unigenes were annotated to 385 metabolic pathway classes, of which 2927 unigenes were involved in carbohydrate metabolism. *Lactiplantibacillus fermentum*, *Lactiplantibacillus plantarum*, and *Lactiplantibacillus brucei* were involved in the metabolism of most nutrients and flavor substances in suansun. Overall, these results provide insights into the suansun microbiota and shed light on the fermentation processes carried out by complex microbial communities.

**Keywords:** suansun; metagenomic; microbial diversity; functional gene; KEGG pathway





## 1. Introduction

Suansun, or sour bamboo shoot, is a traditional Chinese fermented vegetable-based food made from fresh bamboo shoots and fermented naturally without salt. Due to its good storage characteristics and unique flavor after soaking and fermentation, suansun is popular with consumers. Fermented foods are an important part of the daily diet, and fermentation is used to produce and preserve food for extended periods. Several microorganisms and enzymes are involved in the fermentation process, resulting in physiological and biochemical changes in food that are causally related to consumer health [1].

Unlike Korean salt-pickled vegetables such as kimchi, which require much salt [2], suansun does not contain nitrites, which can be harmful to humans. Suansun is rich in dietary fiber and essential amino acids and has been shown to lower blood cholesterol and strengthen the immune system [3,4]. The lactic acid bacteria in suansun not only have probiotic functions but can also produce rich volatile flavor substances, organic acids, bacteriocins, etc., greatly improving palatability [5]. Common diseases such as hypertension, hyperlipidemia, and diabetes, as well as cardiovascular and cerebrovascular diseases, are attracting increasing attention as people's living standards improve. People are increasingly inclined to consume healthy foods that contain probiotics, and the market for suansun is gradually expanding.

Many studies have been conducted on microorganisms, probiotics, and other substances in fermented foods such as wine [6], *Siniperca chuatsi* [7], soybean [8], and cheese [9].

Researchers have sought to understand the microbiome of fermented foods and how it might affect consumers. However, suansun is still mainly produced in small workshops in thousands of households in China, where it is primarily produced using primitive means of production and with varying tastes [10]. Although it is known that it contains a large number of lactic acid bacteria and other microorganisms, the specific microbial community and dominant core flora of unsalted suansun remain unknown; additionally, information on how the suansun microbiome participates in metabolism is scarce.

Here, we applied high-throughput metagenomic sequencing technology to explore the microbial community structure, diversity, and functional genes in suansun from the main production areas in China, information critical for the industrialization of suansun.

## 2. Materials and Methods

### 2.1. Preparation of Suansun and Sampling

Suansun produced from *Dendrocalamus latiflorus* Munro was sampled from farmers in six cities in Guangxi Province, China (Figure 1A). Ten milliliters of fermentation liquid was extracted from the upper, middle, and lower layers of the fermentation vat, mixed, and stored in liquid nitrogen. Sampling was replicated three times. The thawed fermentation liquid was centrifuged at 10,000 rpm for 10 min at 4 °C to harvest microorganisms in the laboratory.

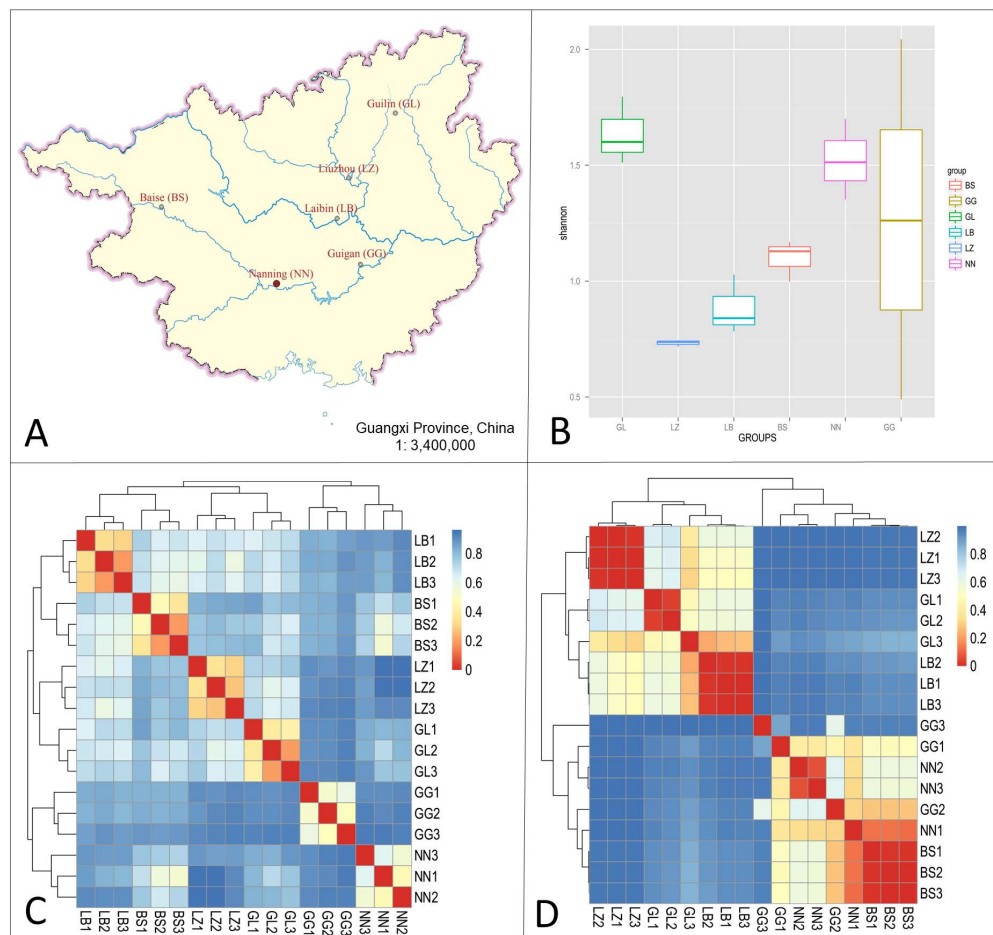

**Figure 1.** Suansun sampling location and microbial diversity. (**A**) Suansun sampling location distribution. The map came from the Ministry of Natural Resources of the People's Republic of China, and the approval number is GS (2019)3333. (**B**) Distribution map of diversity Shannon index between groups. (**C**) Community distance heat map by Jaccard. (**D**) Community distance heat map by thetaYC.

### 2.2. DNA Extraction and Illumina MiSeq Sequencing

A 30 mL sample of suansun fermentation broth was removed from the liquid nitrogen, thawed, transferred into a 50 mL centrifuge tube, and centrifuged at 12,000 rpm for 10 min at 4 °C in a high-speed centrifuge. The supernatant was discarded, the precipitate was washed with PBS, and the microorganisms were collected in a 1.5 mL EP tube for total DNA extraction. The total DNA of all microorganisms in the samples was extracted using the Amgen Plant Genome Extraction Kit (Amgen, Thousand Oaks, CA, USA). The samples were resolved using 0.8% agarose gel electrophoresis (voltage 100 V, time 40 min) to check for degradation and impurities, and quality was determined. The total DNA mass concentration was accurately quantified using Qubit (DNA mass concentration $\geq$ 30 ng/μL, DNA mass $\geq$ 3 μg) to ensure that the second-generation sequencing library construction requirements were met.

The tested DNA samples were randomly broken into 300 bp fragments using an ultrasonic crusher, and library construction was completed by end repair, the addition of A-tail and sequencing junction, purification using gel method, and PCR amplification. The Illumina TruSeq Stranded mRNA LT Sample Prep Kit (Illumina, San Diego, CA, USA) was used for RNA fragmentation, cDNA synthesis, cDNA library construction, PCR amplification of DNA fragments with linkers, and library fragment selection and purification according to the manufacturer's instructions. The libraries were screened using an Agilent Bioanalyzer 2100 (Agilent Technologies, Massy, France) and screened for linker-free sequences with a single peak. The library was also initially quantified using Qubit 2.0, diluted to 2 ng/μL, and the real-time fluorescence nucleic acid amplification detection system (qPCR) method was used to quantify the library.

Amplicons were subjected to paired-end sequencing on the Illumina MiSeq sequencing platform using the PE300 chemical at Ori-Gene Technology Co., Ltd. (Beijing, China). The raw reads were deposited in the NCBI Sequence Read Archive (SRA) database (Accession Number: SRP274288).

### 2.3. Amplicon Sequence Processing and Analysis

After demultiplexing, the resulting sequences were merged with FLASH (version 1.2.11) [11] and quality filtered using fastp (0.19.6) [12]. Afterward, the high-quality sequences were denoised using the DADA2 [13] plugin in the Qiime2 [14] (version 2020.2) pipeline using the recommended parameters, obtaining single-nucleotide resolution based on error profiles within the samples. DADA2-denoised sequences are usually referred to as amplicon sequence variants (ASVs). To minimize the effects of sequencing depth on alpha and beta diversity measures, the number of sequences from each sample was rarefied to 4000, which still yielded an average Good's coverage of 97.90%.

### 2.4. Analysis of Alpha Diversity in Microbial Community

Alpha diversity includes community richness, community evenness, and community diversity. The alpha diversity index was calculated using Mothur [15]. The larger the Shannon's index, the higher the microbial diversity.

$$H_{shannon} = -\sum_{i=1}^{Sobs} \pi_i ln \pi_i$$

*Sobs*: the number of species observed.
$\pi_i$: relative abundance of each species.

### 2.5. Analysis of Beta Diversity in Microbial Community

Jaccard [16] and thetaYC [17] were used to calculate the community distance between the samples and generate the distance matrix.

$$D_{Jaccard} = \frac{S_{AB}}{S_A + S_B - S_{AB}}$$

$S_{AB}$: the number of species shared by community $A + B$.
$S_A$: the number of species in community $A$.
$S_B$: the number of species in community $B$.

$$D_{\theta YC} = 1 - \frac{\sum_{i=1}^{S_T} a_i b_i}{\sum_{i=1}^{S_T} (a_i - b_i)^2 + \sum_{i=1}^{S_T} a_i b_i}$$

$S_T$: the number of species shared by community $A + B$.
$a_i$: the relative abundance of the $i$-th species in community $A$.
$b_i$: the relative abundance of the $i$-th species in community $B$.

### 3. Results and Discussion

#### 3.1. Statistics of Sequencing Data

Compared with 16S sequencing, which has a sequence read length of approximately 200 bp and an effective data volume between 10 and 70 M [18], the metagenomic sequencing in the present study generated a total of 58.18 G of effective data (an average of 9.69 G per sample), which had a higher library coverage and captured more microorganisms in the samples. The average read length of all samples sequenced was 141.2 bp, and the effective read length was 68.75 M, accounting for 91.5% of the original reads; the effective base number was 9.697 G, accounting for 85.9% of the original base number (Table 1).

**Table 1.** Metagenomic sequencing results of different regions' suansun.

| Samples | Original Reads (M) | Effective Reads (M) | Original Bases (G) | Effective Bases (G) | Average Length (bp) |
|---|---|---|---|---|---|
| GL | 71.92 | 66.82 | 10.78 | 9.54 | 143.6 |
| LZ | 67.06 | 62.05 | 10.06 | 8.91 | 143.7 |
| LB | 78.58 | 69.78 | 11.37 | 10.04 | 143.8 |
| BS | 84.85 | 76.03 | 12.72 | 10.56 | 139.4 |
| NN | 77.59 | 70.72 | 11.63 | 9.79 | 138.2 |
| GG | 73.30 | 67.21 | 10.99 | 9.32 | 138.2 |
| Average | 75.10 | 68.75 | 11.27 | 9.70 | 141.2 |

The effective read lengths were iteratively spliced using SOAPdenovo2 [19] to obtain 66,157 scaffolds. The obtained scaffolds were used for gene prediction using MetaGene-Mark [20]. The predicted gene sequences were clustered using CD-HIT [21] software (95% similarity, 90% coverage) to obtain a set of 54,416 unigenes for subsequent bioinformatics analysis.

#### 3.2. Microbial Diversity in Suansun

A total of 156 species of microorganisms were identified from 18 suansun samples from the six main producing areas, belonging to 8 phyla, 16 classes, 30 orders, 63 families, and 92 genera. *Lactobacillales* was the most dominant order among the suansun microorganisms, followed by *Pseudomonadales* (Figure 2). Colony distribution maps for other classification levels are provided in Supplementary Materials. More than 98.86% of the bacterial genes in all samples were from four major phyla, namely Firmicutes, Proteobacteria, Bacteroidetes, and Actinobacteria, with 86.10% from Sclerotinia. The species with an abundance greater than 1% at the phylum level were Firmicutes, Proteobacteria, Bacteroidetes, and Actinobacteria.

At the class level, the species with the highest abundance were Bacilli and *Gammaproteobacteria*. The orders of bacteria with more than 1% abundance included *Lactiplantibacillus*,

*Pseudomonadales*, and *Bacteroidales*. Pseudomonas abundance was significantly higher in suansun produced in Nanning than in other regions.

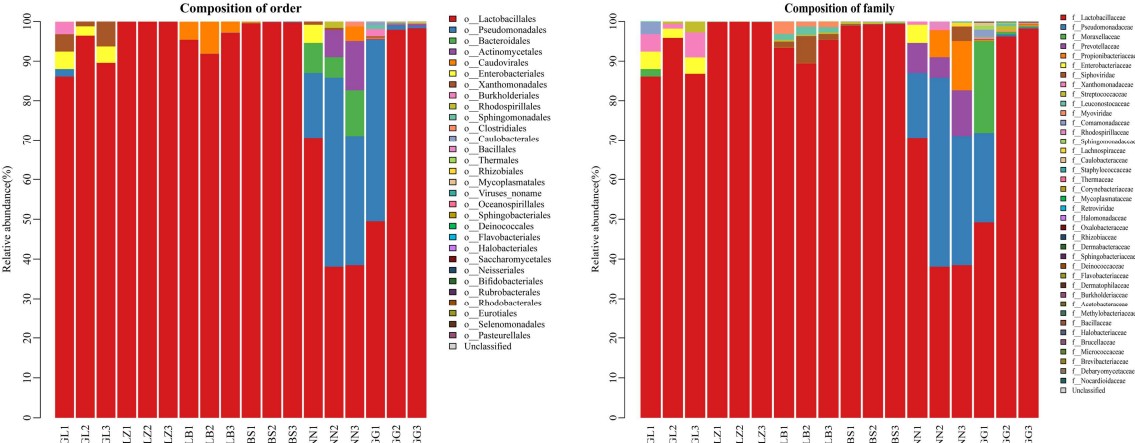

**Figure 2.** Order and family-level colony distribution map. GL, LZ, LB, BS, NN, GG represent Guilin, Liuzhou, Laibin, Baise, Nanning, and Guigang, respectively.

The number of microbial species observed in the samples from Guigang was as high as 77, which significantly differed from other regions ($p < 0.05$) (Table 2). Alpha diversity mainly focuses on the number of species in a local uniform habitat; therefore, it is also called within-habitat diversity. The highest uniformity index (0.69) was obtained in the Nanning area, indicating that the microbial matter was more uniformly distributed in the suansun from this area. Beta diversity refers to the dissimilarity in species composition between different habitat communities along the environmental gradient or species replacement rate along the environmental gradient. This is also called between-habitat diversity (Figure 1C,D). The microbial communities of suansun from Guilin, Liuzhou, and Laibin were between 0.44 and 0.57. The analysis of the distribution of the relative abundance of microorganisms showed that *Lactiplantibacillus fermentum* was the most abundant, indicating that the bacterial communities originating from these three areas were more similar in structure. The microbial community distance of suansun from Baise and Nanning was 0.7, and *Lactiplantibacillus plantarum* was the most abundant. The community distance of suansun originating from Guigang and the other five areas was considerable, with *Lactiplantibacillus brucei* being the main species, indicating that the bacterial community structure of suansun from Guigang and the other five areas differed considerably.

**Table 2.** Bacterial diversity index, evenness index, and abundance of different samples.

| Sampling Location | Number of Microbial Species | Evenness | Shannon Index |
|---|---|---|---|
| GL | 9.67 ± 1.06 d | 0.72 ± 0.06 a | 1.64 ± 0.15 a |
| LZ | 16.33 ± 1.53 c | 0.26 ± 0.00 c | 0.73 ± 0.01 b |
| LB | 25.33 ± 1.53 b | 0.27 ± 0.04 c | 0.88 ± 0.13 b |
| BS | 17.00 ± 4.36 c | 0.39 ± 0.06 b | 1.10 ± 0.09 b |
| NN | 9.00 ± 1.00 d | 0.69 ± 0.04 a | 1.52 ± 0.17 a |
| GG | 77.00 ± 0.90 a | 0.38 ± 0.12 bc | 1.65 ± 0.55 a |

Note: the data in the table are mean ± SE. Columns with different letters are significantly different ($\alpha = 0.05$ by Duncan's test).

### 3.3. Main Active Microorganisms in Suansun

Suansun contains a large number of active microorganisms (Table 3). Most of these active microorganisms are involved in the immunomodulation and bacteriostatic functions of the body. *Lactiplantibacillus boulardii*, *Lactiplantibacillus casei*, and Weisseria esophagus

have antioxidant properties, and *Lactiplantibacillus fermentum*, *Lactiplantibacillus boulardii*, *Lactiplantibacillus pentosus*, and *Lactiplantibacillus pyocephalus* have cholesterol-lowering properties, which help alleviate hyperlipidemia; *Lactiplantibacillus plantarum* and *Lactococcus lactis* are crucial in maintaining intestinal flora and protecting the intestinal mucosa.

**Table 3.** Major active microorganisms in suansun.

| Major Strains | Function |
| --- | --- |
| *Lactiplantibacillus fermentum* | Antibacterial activity, cholesterol-lowering ability, immune activity [22]. |
| *Lactiplantibacillus plantarum* | Immunomodulating effect, inhibit pathogenic bacteria, lower serum cholesterol [23]. |
| *Lactiplantibacillus buchneri* | Produces mannitol, bacteriocins, de-cholesterolization, antioxidant capacity [24]. |
| *Lactiplantibacillus brevis* | High acid production capacity and detoxification, antibacterial, improve the immunity of the body, and other functional characteristics [25]. |
| *Lactiplantibacillus casei paracasei* | Regulates the abundance and proportion of gut flora, protects the liver, and prevents liver damage. [26]. |
| *Lactiplantibacillus pentosus* | Synthesis of extracellular polysaccharides, antitumor, anti-ulcer, immunomodulation, and cholesterol-lowering [27]. |
| *Lactococcus lactis* | Limit intestinal damage and protect the intestinal mucosal barrier [28]. |
| *Weissella cibaria* | Antioxidant activity, inhibition of bacteria [29]. |
| *Leuconostoc citreum* | Production of bacteriocins with broad-spectrum antibacterial action [30]. |
| *Acinetobacter johnsonii* | Improve obesity, lower body fat, and reduce cholesterol [31]. |

A variety of beneficial microorganisms exist in suansun, such as *Pediococcus pentosaceus*, which can improve immune function; *Leuconostoc citreum*, which produces bacteriocins with broad-spectrum antibacterial effects [32,33]; *Alistipes putredinis*, which can regulate human intestinal flora; and Prevotella, which can degrade starch and protein [34] while having probiotic functions. Meanwhile, the dietary fiber in suansun is an ideal nutrient for intestinal flora. The short-chain fatty acids generated by bacterial digestion of fiber nourish the intestinal barrier, improve immune function, help prevent inflammation, and lower tumor development risk [35]. The probiotic nature of suansun is demonstrated by lactobacilli metabolism, which produces substances such as organic acids, hydrogen peroxide, and bacteriocins that can promote nutrient absorption, inhibit pathogens, and are potent antioxidants.

*3.4. Functional Annotation and Analysis of Microbial Gene of Suansun*

The prediction of genes encoding proteins and filtering out coding frame results of less than 100 bp yielded 75,681 genes with a total length of 35.67 Mbp and an average length of 471 bp. The sample genes were de-redundant to yield 54,416 unigenes with a total length of 31.42 Mbp and an average length of 577 bp. The unigenes were matched with common public databases, such as NR, eggNOG, GO, and KEGG. A total of 53,265 unigenes were matched to known protein sequences, with an annotation ratio of 97.88%, and 1151 unigenes did not match the corresponding protein sequences, which might be unique to genes in suansun microorganisms (Table 4).

A total of 37,746 unigenes were compared with the eggNOG database (Figure 3) and annotated to 22 eggNOG functional categories. Information storage and processing had the highest number of genes enriched in replication, recombination, and repair and the lowest number of genes associated with RNA processing and modification classification. Metabolism process classification genes are the most direct functional classification associated with the formation of suansun flavor substances. The number of genes in the eight metabolism categories annotated reached over 10,640, and the top three genes were carbohydrate and metabolism (carbohydrate metabolism, 3029 genes), amino acid transport and metabolism (amino acid metabolism, 2781 genes), and energy production and conversion (energy utilization, 1688 genes), indicating that carbohydrate and amino acid metabolism are essential processes in sour bamboo shoot fermentation.

**Table 4.** Unigenes annotated statistics.

| Functional Database | Number of Unigenes | Percentage |
|---|---|---|
| NR | 39,294 | 72.21% |
| eggNOG | 37,746 | 69.37% |
| KEGG | 19,051 | 35.01% |
| KEGG pathway | 12,751 | 23.43% |
| KO | 14,075 | 25.87% |
| GO | 29,889 | 54.93% |
| Total number of genes | 53,265 | 97.88% |
| Total number of unigenes | 54,415 | 100% |

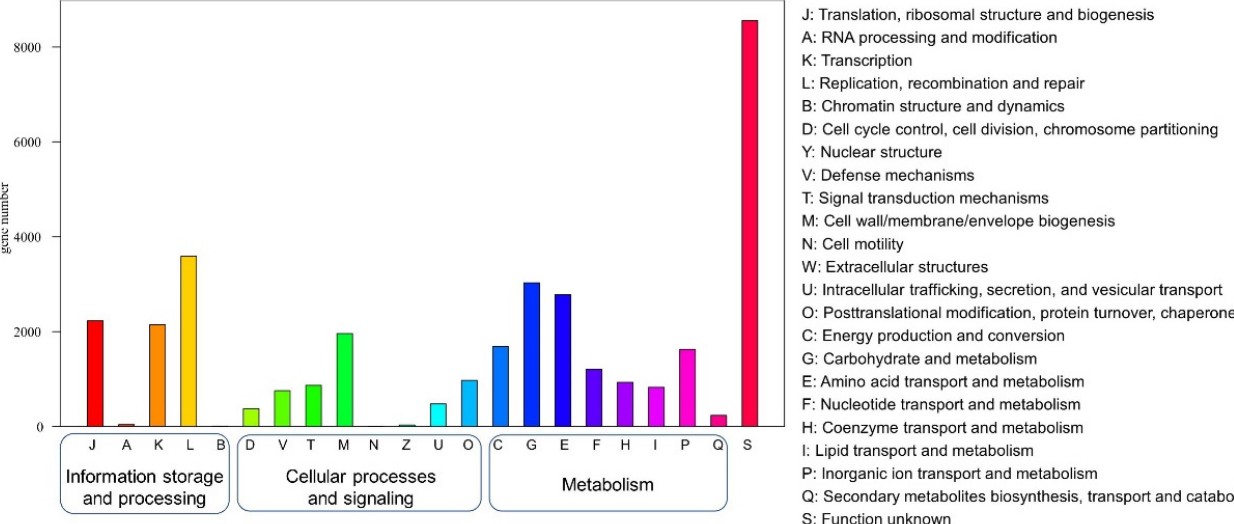

**Figure 3.** Functional classification of the microbial functional gene eggNOG in suansun.

A total of 384 pathways were annotated in the Kyoto Encyclopedia of Genes and Genomes (KEGG) database. The highest number of genes were related to metabolism, followed by genetic information processing, while the lowest number of genes related to cellular processes indicated the high metabolic activity of microorganisms during the suansun fermentation process (Figure 4). The bacteria annotated to the KEGG metabolic pathway in suansun mainly included *Lactiplantibacillus plantarum*, *Lactiplantibacillus pentosus*, *Lactococcus lactis*, *Lactococcus pentosus*, and *Lactococcus weissella*. *Lactiplantibacillus plantarum* was annotated to the KEGG pathway the most, in several pathways, including the tricarboxylic acid cycle (ko00020), pentose phosphate pathway (ko00030), amino acid and nucleotide sugar metabolism (ko00520), and folate synthesis (ko00790). It can be seen that *Lactiplantibacillus plantarum* was one of the more metabolically active microorganisms in suansun.

Based on the GO database, functional annotations indicated 58 GO function categories (Figure 5). The biological process included 22 branches, and the number of genes annotated in single-organism process, cellular process, and metabolic process ranked in the top three, with 16,737 (29.12%), 16,325 (28.41%), and 15,682 (27.29%), respectively. This is consistent with the large number of metabolism-related genes annotated in the eggNOG and KEGG databases. The cellular component included 21 functional categories, and 24,714 genes (54.00%) were annotated to cell and cell parts. Catalytic activity had the highest number of annotated genes in the molecular function. Interestingly, we also identified a number of genes associated with human diseases. As suansun is artificially salted, associated bacterial contamination may be introduced during the preparation process.

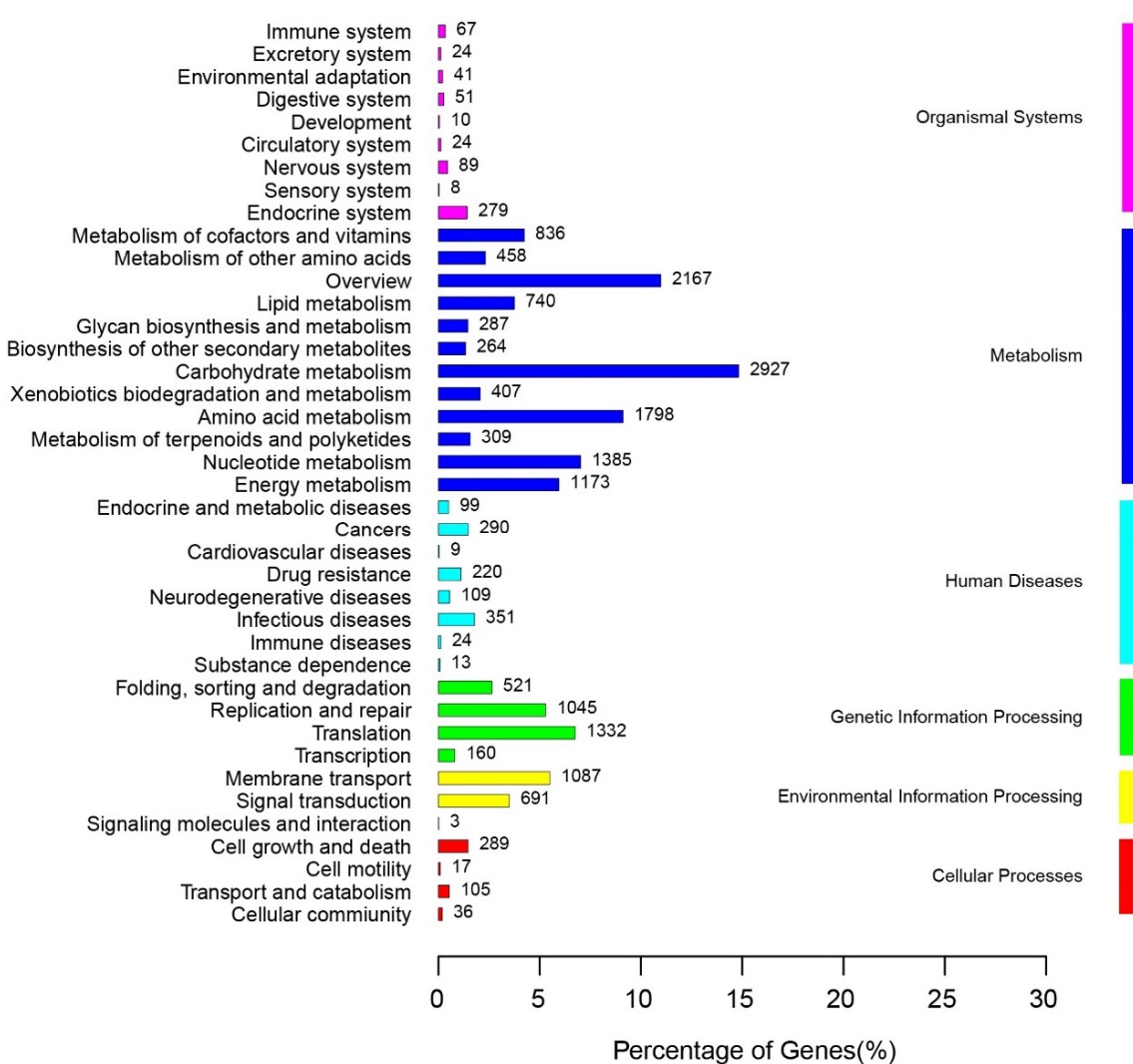

**Figure 4.** KEGG functional classification of functional genes in suansun microorganisms.

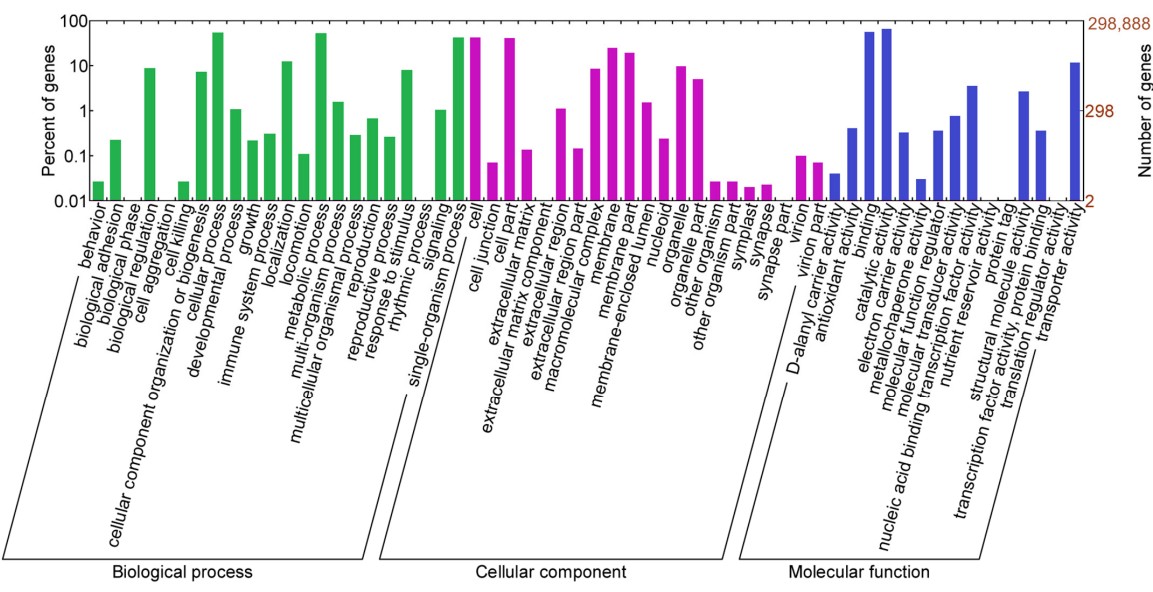

**Figure 5.** GO functional classification of functional genes in suansun microorganisms.

*3.5. Metabolic Pathways of Nutrients and Flavor Substances in Suansun*

Suansun has a unique flavor that exhibits acidity, freshness, and excitement characteristics. The proteins and carbohydrates in suansun are naturally fermented by microorganisms, producing organic acids, volatile odors, and bioactive components that affect the sensory qualities, nutrition, and safety of suansun. At present, the metabolic relationship between microorganisms and flavor substances is still unclear, and the screening of core functional strains for suansun fermentation remains a major challenge. Based on the functional annotation of non-redundant genes, the primary nutrients' and flavor substances' metabolic pathways can be constructed by reference to the KEGG analysis results. The genes, enzymes, and microorganisms in the key metabolic pathways were analyzed in detail to reveal the association between microorganisms and flavor at the genetic level, providing an important reference for the formation of the main flavor substances of suansun.

A total of 13 pathways related to amino acid metabolism were annotated to microbial functional genes in suansun, covering 15 major microbial species (Table 5). Among them, *Lactiplantibacillus plantarum* was annotated to the highest number of KEGG amino acid metabolic pathways, with 13. *Lactiplantibacillus fermentum*, *Lactiplantibacillus plantarum*, *Lactiplantibacillus brucei*, and *Lactiplantibacillus plantarum* were involved in the metabolic processes of most of the nutrients and flavor substances of suansun, which were the key strains screened in this study and could provide some reference for the preparation of food fermentation bacteria.

**Table 5.** Microorganisms associated with amino acid metabolism and their KEGG annotation pathway in suansun.

| Ko Number | KEGG Pathway | Gene Number | Major Microorganisms Annotated to KEGG Pathway |
|---|---|---|---|
| ko00250 | Alanine, aspartate, and glutamate metabolism | 320 | *Lactiplantibacillus plantarum, Lactiplantibacillus pentosus, Pediococcus pentosaceus, Lactiplantibacillus, Lactococcus Lactis, Lactiplantibacillus buchneri* |
| ko00270 | Cysteine and methionine metabolism | 230 | *Lactiplantibacillus plantarum, Lactiplantibacillus parafarraginis, Lactiplantibacillus fermentum, Lactiplantibacillus mucosae, Acinetobacter* |
| ko00260 | Glycine, serine, and threonine metabolism | 220 | *Lactiplantibacillus amylolyticus, Lactiplantibacillus casei, Lactiplantibacillus buchneri, Lactiplantibacillus plantarum, Lactiplantibacillus fermentum, Lactiplantibacillus vaginalis* |
| ko00300 | Lysine biosynthesis | 212 | *Lactiplantibacillus amylolyticus, Lactiplantibacillus casei, Lactiplantibacillus buchneri, Lactiplantibacillus plantarum, Lactiplantibacillus fermentum, Lactiplantibacillus vaginalis, Acinetobacter, Flavobacterium, Acinetobacter johnsonii* |
| ko00400 | Phenylalanine, tyrosine, and tryptophan biosynthesis | 139 | *Lactiplantibacillus buchneri, Lactiplantibacillus amylolyticus, Lactiplantibacillus plantarum, Lactiplantibacillus pentosus, Acinetobacter guillouia, Lactiplantibacillus mucosae* |
| ko00350 | Tyrosine metabolism | 110 | *Lactiplantibacillus buchneri, Lactiplantibacillus amylolyticus, Lactiplantibacillus plantarum, Lactiplantibacillus pentosus, Lactiplantibacillus mucosae, Lactiplantibacillus hilgardii,* |
| ko00340 | Histidine metabolism | 93 | *Lactiplantibacillus buchneri, Lactiplantibacillus plantarum, Lactiplantibacillus parafarraginis, Lactiplantibacillus fermentum, Lactiplantibacillus mucosae, Acinetobacter* |
| ko00330 | Arginine and proline metabolism | 77 | *Pediococcus pentosaceus, Lactiplantibacillus plantarum, Lactiplantibacillus brevis, Lactococcus lactis* |
| ko00280 | Valine, leucine, and isoleucine degradation | 70 | *Lactiplantibacillus pentosus, Lactiplantibacillus casei, Lactiplantibacillus buchneri, Lactiplantibacillus plantarum* |
| ko00471 | D-glutamine and D-glutamine metabolism | 55 | *Lactococcus lactis, Lactiplantibacillus buchneri, Acinetobacter parvus, Lactiplantibacillus plantarum, Lactiplantibacillus pentosus* |
| ko00473 | D-alanine metabolism | 51 | *Lactococcus weissella, Lactiplantibacillus, Lactiplantibacillus fermentum, Lactococcus lactis, Lactiplantibacillus plantarum, Lactiplantibacillus pentosus* |
| ko00380 | Tryptophan metabolism | 45 | *Lactiplantibacillus buchneri, Lactiplantibacillus amylolyticus, Lactiplantibacillus plantarum, Lactiplantibacillus pentosus, Acinetobacter guillouia, Lactiplantibacillus mucosae* |
| ko00290 | Valine, leucine, and isoleucine biosynthesis | 38 | *Lactiplantibacillus buchneri, Lactiplantibacillus amylolyticus, Lactiplantibacillus plantarum, Lactiplantibacillus pentosus* |

## 4. Conclusions

Here, high-throughput sequencing technology was used to investigate the metagenomics of suansun, a traditional fermented food from Guangxi, the main production area of China. The community structure, functional genes, and metabolic pathways of microorganisms in suansun were revealed, and the association between microorganisms and amino acids in suansun was explored to clarify the connection between microorganisms and suansun flavor at the genetic level.

A total of 156 microorganisms belonging to 8 phyla, 16 classes, 30 orders, 63 families, and 92 genera were detected in suansun. *Lactiplantibacillus fermentum*, *Lactiplantibacillus plantarum*, *Lactiplantibacillus amyloliquefaciens*, *Lactiplantibacillus brucei*, and *Lactiplantibacillus brevis* were the dominant species in suansun. A total of 75,681 genes, 54,416 unigenes, and 19,051 functional genes were annotated to 384 metabolic pathways among suansun microorganisms, of which 72 pathways were involved in metabolism. *Lactiplantibacillus fermentum*, *Lactiplantibacillus plantarum*, *Lactiplantibacillus brucei*, and *Lactiplantibacillus plantarum* play crucial roles in suansun fermentation. This study provides a theoretical basis for understanding the discovery and utilization of suansun as a health food.

However, the sour bamboo shoot fermentation process is complex, and many aspects of the changes occurring are likely not caused by microorganisms. It is necessary to conduct in-depth studies on the changes in its quality and flavor caused by non-microbial factors using metabolomics and proteomics techniques in the future.

**Supplementary Materials:** The following are available online at https://www.mdpi.com/article/10.3390/pr9091669/s1, Figure S1: Class level colony distribution map, Figure S2: Genus level colony distribution map, Figure S3: Phylum level colony distribution map.

**Author Contributions:** Data curation, Y.H. and X.C.; formal analysis, J.Z. and W.J.; funding acquisition, Q.G.; writing—original draft, Y.H.; writing—review and editing, Q.G. All authors have read and agreed to the published version of the manuscript.

**Funding:** This work was supported by research projects of the Talent Introduction Project Study of Nanjing Forestry University on Ginkgo biloba and other important tree germplasm resources (GXL2018001).

**Institutional Review Board Statement:** Not applicable.

**Informed Consent Statement:** Not applicable.

**Acknowledgments:** We thank the Co-Innovation Center for Sustainable Forestry in Southern China and the International Center of Bamboo and Rattan for allowing us to use their research facilities.

**Conflicts of Interest:** The authors declare no conflict of interest.

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
