# Peer review of "Metagenomic Analysis of Suansun, a Traditional Chinese Unsalted Fermented Food"

_processes, doi:10.3390/pr9091669_

Round 1

Reviewer 1 Report

Author present metagenomic analysis of traditional fermented product microbiota. The technical and methodological part is well presented, but results and discussion section needs improvements. Discussion is to general related to properties (probiotic, aroma..) of microbiota (species) involved in fermentation of products. Nomenclature of LABs should be up-to-date, link is attached. All specific comments are given in attached text.

Author Response

Thank you for your detailed comments and suggestions.

We have made one by one changes in the text according to your comments. We have revised the latest nomenclature of Lactobacillus species. Further, we change the description of "probiotics". Changes to the details of the manuscript have been marked in the text. Please review the details of the changes already marked in the manuscript. 

Thanks again!

Reviewer 2 Report

Presented paper is very interesting, because Authors described microbiome of suansun, a traditional Chinese unsalted fermented food. However, I would recommend describing the preparation of DNA libraries in more detail, as there is no information about the stage of preparing samples for sequencing. The second important thing is definition of probiotics. When we say about microbiome of fermented food, we can't say that these bacteria are probiotic. These are lactic acid bacteria (LAB), but not probiotics, because probiotic strains are well-described bacteria that have well-proven scientific pro-health effects. LABs presented in fermented food are just bacteria that replenish our microbiome with a likely positive effect on our body.

Author Response

Thank you for your comments.

Based on your comments, we have detailed the description of  the preparation of DNA libraries. We have revised the description of "probiotics" in the text to avoid ambiguity among readers. In addition, we have made other further revisions based on the comments of other reviewers. Please see the labeled section in the manuscript for details.

Thanks again!

Round 2

Reviewer 1 Report

Dear authors, please again check the nomenclature. Your changes in manuscript are not in line with paper I recommended. When search for new names, please check each species mentioned in your paper, for example note Lactobacillus plantarum (old name)_ now is Lactiplantibacillus plantarum. When use abbreviated genus name, some differences should be visible between them; for example Lactobacillus can be Lb., Leuconostoc Ln., Lactococcus Lc., Lactiplantibacillus Lpb.; please check other literature reffering this abbreviations and names of LAB species.

Here is also one tool: http://lactotax.embl.de/wuyts/lactotax/

Also when discussing properties of Lactiplantibacillus plantarum, consider to quote this paper: https://www.hindawi.com/journals/bmri/2018/9361614/

Author Response

Dear reviewer,

Thank you for the support.

We have once again carefully revised the latest naming rules.

Please see the section marked in the text.
